# Biomedical Named Entity Recognition via Reference-Set Augmented Bootstrapping

## Abstract

We present a weakly-supervised data augmentation approach to improve Named Entity Recognition (NER) in a challenging domain: extracting biomedical entities (e.g., proteins) from the scientific literature. First, we train a neural NER (NNER) model over a small seed of fully-labeled examples. Second, we use a reference set of entity names (e.g., proteins in UniProt) to identify entity mentions with high precision, but low recall, on an unlabeled corpus. Third, we use the NNER model to assign weak labels to the corpus. Finally, we retrain our NNER model iteratively over the augmented training set, including the seed, the reference-set examples, and the weakly-labeled examples, which results in refined labels. We show empirically that this augmented bootstrapping process significantly improves NER performance, and discuss the factors impacting the efficacy of the approach.

## 1 Introduction

The increasing wealth of available data fuels numerous machine learning applications. Unfortunately, much of this data is unlabeled, unstructured and noisy. Supervised learning achieves the best task performance, but obtaining training labels is expensive. Crowd-sourcing could provide labels at scale, but may not be feasible for acquiring high-quality labels in technical domains, such as biomedicine that requires expert annotators. In this paper, we explore augmented bootstrapping methods that leverage automatically assigned noisy labels obtained from a large unlabeled corpus.

The biomedical literature is a high-impact domain with scarce annotations. Unlocking the knowledge in this data requires machine reading systems that automatically extract important concepts in the text, such as entities and their relations. A critical component of such systems is reliable Named Entity Recognition (NER), which aims to identify parts of the text that refer to a named entity (e.g., a protein). In line with advancements in many domains, most state-of-the-art NER approaches use a deep neural network model that relies on a large labeled training set, which is not usually available in biomedical domains. To address label scarcity, we propose a framework to train any effective neural NER model by leveraging partially labeled data. We do this by creating an augmented training set using a small fully-labeled *seed* set, and an unlabeled *corpus* set, which we weakly and automatically label, and then refine its labels via an iterative process.

Our main **contributions** include: (1) An augmented bootstrapping approach combining information from a reference set with iterative refinements of soft labels to improve NER in a challenging domain (biomedicine) where labelling is expensive. (2) A detailed analysis in a controlled setting to study different aspects affecting performance. (3) An analysis of reference-based automated approaches to labeling data, showing that naive labeling decreases performance and how to overcome it.

## 2 Related Work

Many effective NER systems assume a fully-supervised setting to train a neural network model (Liu et al., 2018; Ma & Hovy, 2016; Lample et al., 2016). Recently, distant supervision has been applied to language-related tasks such as phrase mining (Shang et al., 2018a), relation extraction (Mintz et al., 2009), and entity extraction (He, 2017). For NER, Fries et al. (2017) automatically generated candidate annotations on an unlabeled dataset using weak labellers. Ren et al. (2015) and He (2017) used knowledge bases and linguistic features to tag entities. Our approach combines knowledge extracted from an external reference set with noisy predicted labels and refines them an iteratively.

Using a reference set Ratner et al. (2017) proposed heuristic-based functions to label data with low accuracy. Shang et al. (2018a;b) described techniques to automatically tag phrases based on

knowledge bases such as MeSH and CTD in the biomedical domain. However, in NER systems with weak supervision, wrongly-labeled entities negatively affects the overall performance (Shang et al., 2018b). We show that our proposed iterative training technique is able to make the learning process more robust to noisy labels.

Our method is closely related to bootstrapping approaches. Yarowsky (1995) introduced the bootstrapping technique by training a tree-based classifier for word-sense disambiguation on labeled seed data and then using it to predict on an unlabeled corpus which further is used for training the model iteratively until convergence. Later Kozareva (2006) bootstrapped statistical classifiers for NER. Abney (2004) and Haffari & Sarkar (2007) applied bootstrapping for language processing, and Reed et al. (2015) for image classification.

We propose an augmented bootstrapping technique for the state-of-the-art neural NER model applied to biomedical literature. In contrast to the standard bootstrapping techniques that use hard labels, we leverage and refine soft label values, which may be more suitable for noisy data. More importantly, we further augment the bootstrapping process via a simple domain-independent data annotation scheme based on a reference set, which is in contrast to the hand-crafted domain specific rules or the linguistic or morphological characteristics used in standard bootstrapping approaches.

## 3 REFERENCE-SET LABELLING AND AUGMENTED BOOTSTRAPPING

Our main goal in this study is to use easily available external information to leverage unlabeled data and reduce the need for an expensive fully-labeled dataset. We assume to have a small fully-annotated *seed* dataset $\mathcal{D}_s$ that has every token tagged by entity type and a larger *unlabeled corpus* $\mathcal{D}_c$. We seek to automatically generate an augmented dataset by partially, and possibly noisily, labeling $\mathcal{D}_c$. We show that training a (Neural) NER system over the combined seed and augmented datasets achieves the performance of systems trained with an order of magnitude more labels.

### 3.1 LEVERAGING REFERENCE SETS AND ITERATIVE LABEL REFINEMENT

We propose an iterative solution to improve NER by labeling the corpus dataset using two complementary sources of information. First, we train a NER model using the small seed dataset $\mathcal{D}_s$ and use it to label the unlabeled corpus $\mathcal{D}_c$, we call this set of labels *predicted labels*. Second, we use *search policies* over a *reference set* to find mentions of entity names in the unlabeled corpus $\mathcal{D}_c$, we call these set of labels *reference-based labels*. We combine the seed, the predicted and the reference labels to retrain the NER model. We use the updated model to iteratively refine the *predicted labels* portion of the corpus set.

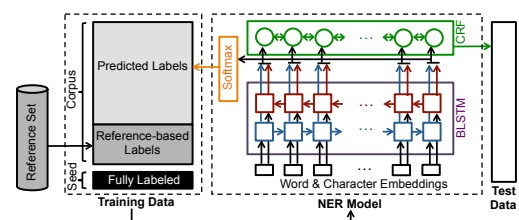

Figure 1: NNER with Augmented Bootstrapping Architecture.

Figure 1 and Algorithm 1 show the overall process of our method. We use soft scores (between 0 and 1) to label the corpus set, instead of the binary labels produced by the CRF layer used in state-of-the-art NER models. Our aim is to let the model iteratively reinforce the weak signals in the soft scores to improve the label quality.

### 3.2 BASE NER MODEL AND SOFT LABELING

Recent high-performing neural NER (NNER) models (Lample et al., 2016; Ma & Hovy, 2016) use Bi-directional LSTM (BiLSTM) layers trained on character and word embeddings. The character embeddings are learned over the training data using a separate BiLSTM layer, and are concatenated with GloVe word embeddings (Pennington et al., 2014). We use an open-source Tensorflow implementation of this model (Genthial, 2017), which achieves state-of-the-art NER performance on the CoNLL 2003[1] dataset. To produce soft scores for each tag in our experiments, we replace the CRF layer with a *softmax* layer. Entities found via the reference set receive a score of 1.

---

[1]https://www.clips.uantwerpen.be/conll2003/ner/

## 4 EXPERIMENTAL ANALYSIS AND RESULTS

We show the effectiveness of our approach in a hard NER problem, extracting protein mentions from the biomedical literature, and systematically evaluate the contribution of the different techniques.

We use the BioCreative VI Bio-ID dataset Arighi et al. (2018), which contains 13,573 annotated figure captions corresponding to 3,658 figures from 570 full length articles from 22 journals, for a total of 102,717 annotations. The Bio-ID dataset is split into a training set of 38,344 sentences, a development set of 4,243 sentences, and a test set with 14,079 sentences. The tokens are tagged using the BIO scheme (Beginning, Inside and Outside of entities).

The Bio-ID dataset provides us with a controlled environment where we can evaluate our methods, since it provides ground truth on the labels. The rationale of the following experiments is to simulate our desired data augmentation scenario, which is to search for sentences containing relevant bioentities (e.g., proteins) in a large corpus, such as PubMed Central. We evaluate our three main techniques, namely (1) using a reference set of entity names (i.e., protein names from UniProt), (2) predicting labels for unknown tokens using a NNER system trained in a small fraction of the data, and (3) refining the label predictions by retraining the NNER system iteratively. We focus on protein/gene annotations for simplicity (51,977 mentions with 5,284 distinct entities).

---

**Algorithm 1:** Assignment algorithm

---

**Function** *IterativeTrain* $(\mathcal{D}_s, \mathcal{D}_c)$
    **Input:** Labeled seed data $(\mathcal{D}_s)$
    **Input:** Unlabeled corpus $(\mathcal{D}_c)$
    **Output:** Iteratively trained model $(\mathcal{M}_K)$
    Train model $\mathcal{M}_0$ on $\mathcal{D}_s$
    **for** *i in 1 . . . K* **do**
        $\mathcal{D}_c^{(i-1)*} \leftarrow$ Predict using $\mathcal{M}_{i-1}$
        $\mathcal{D}_c^{(i-1)} \leftarrow$ Relabel $\mathcal{D}_c^{(i-1)*}$
        **s.t.**
        **if** *token ∈ Reference Set* **then**
            $score_{tag}(token) \leftarrow 1$
        **end**
        Train model $\mathcal{M}_i$ on $\mathcal{D}_s + \mathcal{D}_c^{(i-1)}$
    **end**
    **return** $\mathcal{M}_K$

---

Our experimental evaluation appears in Table 1, which shows Precision, Recall and $F_1$ over the Bio-ID test set for different conditions. Experiments 1 and 2 (rows 1, 2) show results of the NNER system trained over the full Bio-ID training dataset, which on the test set achieves $F_1$ of 82.99% (BiLSTM) and 83.34% (BiLSTM-CRF). This simulates the performance over a large amount of labeled data and is our gold standard upper limit. For the remaining rows, we train a NNER system over a small dataset (3% of the Bio-ID training dataset), which we refer as NNER-3%. We use the NNER-3% model to predict labels for unknown tokens (noisily, since its accuracy is not perfect). Then, we apply different data augmentation techniques over the remaining 97% of the Bio-ID training dataset, which simulates the accessibility of a large unlabeled corpus.

| Experiment Name | Arch | Iters | True Labels | Ref Set | Pred Labels | Init P | Init R | Init F1 | Best P | Best R | Best F1 |
|---|---|---|---|---|---|---|---|---|---|---|---|
| | | | | | | Seed data (3%) | | | Seed + Augmentation | | |
| 1. 100% Training | BiLSTM | 0 | 100% | No | No | - | - | - | 78.73 | 87.73 | 82.99 |
| 2.        ” | + CRF | 0 | 100% | No | No | - | - | - | 80.75 | 86.09 | 83.34 |
| 3. Partial Ref Set, No Iters | BiLSTM | 0 | 40% | No | No | - | - | - | 78.70 | 36.05 | 49.45 |
| 4.        ” | + CRF | 0 | 40% | No | No | - | - | - | 68.51 | 51.31 | 58.67 |
| 5. Partial Ref Set, Iterative | BiLSTM | 10 | 40% | No | Yes | 67.60 | 79.14 | 72.91 | 68.37 | 89.66 | 77.58 |
| 6.        ” | + CRF | 10 | 40% | No | Yes | 67.94 | 86.77 | 76.21 | 72.79 | 88.92 | 79.75 |
| 7. Ref Set, Iterative | BiLSTM | 10 | No | C1 | Yes | 69.71 | 75.96 | 72.70 | 61.60 | 84.71 | 71.33 |
| 8.        ” | BiLSTM | 10 | No | C2 | Yes | 69.71 | 75.96 | 72.70 | 70.30 | 84.23 | 76.63 |
| 9.        ” | + CRF | 10 | No | C2 | Yes | 69.71 | 75.96 | 72.70 | 71.03 | 85.74 | 77.70 |

Table 1: Experimental Evaluation. [C1 = Exact search (P=59.23, R=18.66). C2 = Removed words in English dictionary and words less than 4 characters; case-insensitive search (P=90.20, R=39.35)].

Experiment 3 (row 3 in Table 1) shows the results for a simple baseline where we train our NNER system over the 3% seed combined with one true protein label per sentence for the remaining 97% of the Bio-ID training dataset, which removes ∼60% of the protein labels. This experiment simulates an augmentation method with perfect precision, but a recall of only 40%. Experiment 4 adds the CRF to the architecture over the same scenario, which results on a ∼9 point increase on $F_1$ to reach ∼58% (although precision suffers). Even in this somehow unrealistic scenario that includes many of the available labels, the overall performance is significantly diminished from the the system trained on 100% of the data (∼25 percentage points below in $F_1$).

Experiments 5 and 6 show the effect of our iterative label refinement method. We first train NNER-3% on the seed data. Then we combine the seed, with the perfect precision (but partial) labels as in experiments 3 and 4, and with the noisy predicted labels for the remaining tokens in the (97% of the) training dataset. Surprisingly, training over only 3% of the data already achieves a good $F_1$ of 72.91% for the BiLSTM architecture and 76.21% for the BiLSTM+CRF architecture. When we retrain this base system iteratively, the accuracy of the predicted labels increases, which leads to an improvement of $\sim$3-4 percentage points in $F_1$ (to 77.58% for the BiLSTM and 79.75% for the

Table 2: Performance of iterative refinement (E4,5).

| It | BiLSTM | | | BiLSTM+CRF | | |
|---|---|---|---|---|---|---|
| | P | R | F1 | P | R | F1 |
| 0 | 67.60 | 79.14 | 72.91 | 84.14 | 66.49 | 74.28 |
| 1 | 68.47 | 85.61 | 76.08 | 67.94 | 86.77 | 76.21 |
| 2 | 68.92 | 86.54 | 76.73 | 68.73 | 88.59 | 77.41 |
| 3 | 68.86 | 87.78 | 77.18 | 68.69 | 88.91 | 77.51 |
| 4 | 69.13 | 88.11 | 77.47 | 70.26 | 88.18 | 78.21 |
| 5 | 69.13 | 88.00 | 77.43 | 69.48 | 88.78 | 77.95 |
| 6 | 68.91 | 88.59 | 77.52 | 70.09 | 88.79 | 78.34 |
| 7 | 68.44 | 88.38 | 77.15 | 70.35 | 89.63 | 78.83 |
| 8 | 68.26 | 89.29 | 77.37 | 69.73 | 89.41 | 78.36 |
| 9 | 68.01 | 89.02 | 77.11 | 69.30 | 89.88 | 78.26 |
| 10 | 68.37 | 89.66 | **77.58** | 72.29 | 88.92 | **79.75** |

BiLSTM+CRF). Thus, the iterative label refinement method reduces the distance to the 100% trained system from 25 to 4 percentage points, which is a substantial improvement.

Table 2 shows the evolution of the iterative label refinement procedure. We train NNER-3% (row 0) and use it to predict labels for unknown tokens repeatedly, which yields a jump in performance in the first iteration, since the predicted labels are informative, and then a more gradual improvement as the labels are increasingly refined.

Finally, the remaining experiments simulate the more realistic scenario we seek, where we search for sentences in a large corpus to be labeled automatically. In experiment 7, we simply use our reference set to directly search for exact mentions in the corpus. That is, we search in a case sensitive way for protein/gene names from UniProt in the 97% dataset that represents our large corpus. Matching tokens are labeled as true proteins for training. Since we know the true labels, we can compute the precision (=59.23%) and recall (=18.66%) of this selection technique, which is in fact quite poor. Even using our iterative training technique that produced good results in the previous experiments, somewhat decreases the performance (from $F_1$ =72.70 for NNER-3% down to 71.33%). The low quality of the augmented data introduces too much noise to improve performance.

To lower the noise, we refined our search procedure to improve the precision. For experiments 8 and 9, we filtered the names of our reference set, since after error analysis we discovered that many protein names were ambiguous. For example, the token ANOVA is a name of Q9UNW protein in UniProt, and a well-known statistical procedure. Thus, we removed all protein names that appear in an English dictionary from our search. More drastically, we also removed protein names of less than 3 characters, to avoid capturing acronyms that may not really be protein mentions. Finally, we also relaxed the matching strategy to be case insensitive and also to allow for partial matches. For example, when searching for TIGAR, we will accept "Flag-tagged-TIGAR". This selection techniques yield an improved precision (=90.20%) and recall (=39.35%) on identifying correct proteins in Bio-ID. We then reconstruct our augmented training dataset that combines the seed, reference-set, and the predicted labels by NNER-3% and iterative refinement. Our method achieves a $F_1$ of 76.63% for BiLSTM and of 77.70% for BiLSTM+CRF.

In summary, through these experiments we show that using a small labeled dataset and our automatic data augmentation procedure, we achieve a performance approaching that of a system trained with over 30 times more labeled data.

## 5 CONCLUSION AND FUTURE DIRECTIONS

We proposed a method to improve NER with limited labeled data, which is often the case in technical domains, such as biomedicine. Our method combines bootstrapping and weakly-labeled data augmentation by using a small fully-labeled seed dataset and a large unlabeled corpus, automated labelling using a reference set, and an iterative label refinement process. Our experimental evaluation shows performance equivalent to systems trained with an order of magnitude more labeled data.

In future work, we aim to explore additional augmentation methods over other challenging datasets. We plan to apply the findings of these controlled experiments to a much larger in-the-wild scenario where we use all the available labeled data as the seed and operate over a large corpus (e.g., all of PubMed, PubMed Central) to improve state-of-the-art NER performance.

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
