# OpenReview forum: "Biomedical Named Entity Recognition via Reference-Set Augmented Bootstrapping"
_ICLR.cc/2019/Workshop/LLD — Submitted to LLD 2019_

### Official Review · AnonReviewer1 · 2019-04-07
**Experimental settings are unclear**

**Rating:** 2
**Confidence:** 2

**Review:**

This short paper presents a framework to train a neural Name Entity Recognition (NNER) model, from partially labelled data.
The idea is to first train a NNER model on a small but fully labelled dataset (Ds), and use this model to labelled another, unlabelled, dataset (Dc). A reference set is then used to find mentions of entities in Dc. Finally, both Ds and Dc are combined to retrain the NNER model, which is then used to refine the labeling of Dc.While the second step is about data augmentation, the third and last step is about training the NNER model iteratively. Both steps are repeated K times.
However, it is not explicitly mentioned whether the NNER model is retrained from scratch, or if it is fine-tuned.
The authors should give information regarding the setup of their model since they obviously change it from (Genthial, 2017).
While this part is of the paper  was easy to follow, I found the section about the experiments and the discussion of the results much more confusing. The authors report results from 9 experiments in a single table, with a confusing naming, e.g.:
- what does "+CRF" mean? Does it refer to the architecture from (Genthial, 2017) with the CRF layer replaced with a softmax layer?
- the NNER-3%, why 3% and how did you select this subset of the Bio-ID dataset?
- Experiments 7 to 9, C1 and C2 are not explained, just mentioned in the caption.
Overall, Section 4 is really dense and hard to follow. Instead, the authors should consider splitting the explanation for each of their experiments in individual paragraphs.

---

### Decision · Program_Chairs · 2019-04-16
**Acceptance Decision**

Reject